# How Do Nature-Based Activities Benefit Essential Workers during the COVID-19 Pandemic? The Mediating Effect of Nature Connectedness

**DOI:** 10.3390/ijerph192416501

**Published:** 2022-12-08

**Authors:** Xiang Huang, Liangyi Luo, Xinyi Li, Yingxin Lin, Zhiqiang Chen, Chen Jin

**Affiliations:** 1School of Tourism Management, South China Normal University, Guangzhou 510006, China; 2South China Ecological Civilization Research Center, South China Normal University, Guangzhou 510006, China; 3Faculty of Education, Beijing Normal University, Beijing 100875, China; 4China Nature Education Network, Shenzhen 518028, China

**Keywords:** nature connectedness, five senses, perceived restoration, state anxiety, healing effect

## Abstract

Although many studies have suggested that nature-based activities have a healing effect on human beings, there is little research on the underlying mechanism. This study investigated the role of nature connectedness in the relationship between the perception of nature and individuals’ physical and psychological health. We recruited essential workers who participated in disease prevention and control during the COVID-19 pandemic and their family members as the subjects for this study. The stress levels experienced by this group made them an ideal sample. The results of a survey-based study showed that nature-based activities had a positive effect on alleviating state anxiety levels. The results also showed that nature-based activities affected perceived restoration via the feeling of nature connectedness. This study examined the healing effect of nature-based activities that stimulate the five senses and nature connectedness and explored the potential of nature-based treatments for people experiencing high levels of stress.

## 1. Introduction

During the COVID-19 pandemic, essential workers who engaged in disease prevention and control, such as medical workers, police officers, community officers, and volunteers, faced enormous physical and mental challenges. Their families were worried and anxious about them. Research has indicated that stress makes individuals more vulnerable to mental health problems [1,2]. One study on frontline workers’ stress reported that medical and nursing staff in Wuhan suffered from varying degrees of mental health problems (N = 994) [3]. In another study on professionals’ mental health, 50.7% of the participants reported depressive symptoms, 44.7% reported anxiety, and 36.1% reported sleep disturbances (N = 1563) [4]. Huang and Zhao found that healthcare workers were more likely to have poor sleep quality than other occupational groups [5]. Lu et al. found significant differences in the severity of fear, anxiety, and depression experienced by medical and administrative staff [6]. Hence, it is important to use appropriate methods to improve mental health interventions and reduce the negative effects of stress because the negative effects of high-stress environments do not dissipate immediately after the stressful event is brought under control.

The biophilia hypothesis [7] suggests that human physiology is better adapted to the natural environment than to the urban environment of cities because humans evolved in the natural environment, and nature is obviously attractive to humans, which is why humans enjoy contact with nature. Based on this hypothesis, Frumkin suggested that certain kinds of contact with the natural world are beneficial for health and provided evidence relating to the benefits from animals, plants, landscapes, and the wilderness [8]. Follow-up studies have also shown that contact with nature improves cognition, emotional balance, and health [9,10,11]. Therefore, nature-based activities might be effective in managing high levels of stress.

Nature connectedness, also referred to as nature relatedness, conceptualizes the cognitive and emotional connections with nature, which explain how individuals understand and identify their relationship with nature. This is reflected in an individual’s experience of connectedness with the natural environment [12]. Research has suggested that an individual’s sense of nature connectedness can be improved by actively connecting with nature and that it has physical and psychological benefits, such as recovery from mental fatigue, improved cognitive performance, and enhanced well-being. Howell et al. conducted two studies with undergraduate students and found that nature connectedness has a significant association with well-being [13]. Capaldi et al. [9] also supported the idea that nature connectedness has a positive association with happiness. However, more research is needed to ascertain whether and how nature connectedness promotes physical and psychological health. Berto claimed that restorative environments help maintain and restore the ability to direct attention and that nature has restorative benefits, but he did not determine the role of nature connectedness in this effect [14]. Martyn et al. argued that individuals who connect with nature more frequently have a higher level of nature connectedness and lower anxiety levels, but they did not determine the underlying mechanism [15].

In the context of the COVID-19 pandemic, we selected frontline workers who were engaged in epidemic prevention and control as the subjects to investigate how nature-based activities affect physical and mental health and explore the mechanism of nature connectedness. Our aim was to determine the specific factors of nature-based activities that contribute to healing and to provide a theoretical framework for understanding how nature-based activities enable psychological adjustment for people experiencing stress.

## 2. Literature Review and Hypothesis Development

### 2.1. Theoretical Background

Studies have suggested that nature-based activities have several benefits, such as restoring self-cognition [16], paying more attention to oneself [17], feeling more positive emotions [18], and experiencing less anxiety [19]. Attention restoration theory [20,21] and stress recovery [22] are the two main concepts used to explain the healing effects of nature-based activities. Attention restoration theory posits that concentrating on one task for a long time, even when the task is pleasurable, consumes a large amount of directed attention and leads to mental fatigue. Directed attention plays an important role in cognitive functioning, such as information processing, impulse restraint, and distraction minimization. Making a conscious effort to stay healthy requires directed attention, which consumes considerable energy and leads to fatigue. In contrast, undirected attention is non-referent and spontaneous and does not consume large amounts of an individual’s mental resources. An individual’s capacity for directed attention can be restored after spending some time in an undirected attention mode. The natural environment contains more elements that attract an individual’s undirected attention than the urban environment. Therefore, individuals can have a restorative experience in the natural environment and recover from directed attention fatigue. The concept of stress recovery suggests that the natural environment contains fewer threatening elements than the urban environment, which makes people feel safe. Therefore, the natural environment provides a restorative experience, arouses positive emotional reactions, and reduces stress. During the COVID-19 pandemic, frontline workers had to concentrate on controlling the spread of the virus, which consumed excessive amounts of their directed attention, leading to mental fatigue and mental health problems. Excessive workload, isolation, and discrimination further added to the pressure. This study examined whether guiding affected individuals to participate in nature-based activities helped them recover from the effects of this pressure and directed attention fatigue.

The natural environment and natural elements have been widely used in alternative therapies, such as wilderness therapy, which was developed in America, and forest therapy and horticultural therapy, which are popular in developed countries, such as Japan and Germany. Wilderness therapy was developed to help adolescents with behavioral, mental health, and substance abuse problems. Its main purpose is to assess and treat problem behaviors and cultivate personal and social responsibility among troubled adolescents [23]. Forest therapy, also called “forest bathing” or *shinrin-yoku*, is defined as immersing oneself in the atmosphere of a forest [24]. Its main purpose is to promote physical and psychological health through the healing effect of being in a forest. Although their purposes and forms are different, both forest therapy and horticultural therapy are conducted in natural environments. Horticultural therapy aims to adjust an individual’s social, educational, psychological, and physical state through gardening activities. Unlike wilderness and forest therapies, horticultural therapy is conducted in an artificial environment, such as in a garden or on agricultural land, and is aimed at enhancing sensory stimulation. However, few studies have examined how these alternative therapies work. This study introduces the idea of nature connectedness, perceived restoration, and sensory stimulation to explain the mechanism of the effect of nature-based therapies.

### 2.2. Nature Connectedness

The literature does not provide a unanimous definition of the concept of nature connectedness. Different scholars have used different terms and developed different assessment scales, such as love and care for nature [25], connectivity with nature [26], emotional affinity toward nature [27], dispositional empathy with nature [28], inclusion of nature in the self or inclusion in nature [29], and ecological embeddedness [30].

Scholars have explained nature connectedness from cognitive, emotional, and other perspectives. Schultz defined environmental concern as an individual’s perception of the connection between the self and nature [29]. Accordingly, he argued that the more an individual takes nature into account and the more attention they pay to nature, the greater their environmental concern. Schultz developed the Inclusion of Nature in the Self scale to assess the degree to which an individual includes nature in their self-consciousness [29].

Kals et al. developed the Emotional Affinity Toward Nature scale to measure emotional connection between individuals and nature [27]. They considered emotional affinity toward nature to be an emotional bond that is different from an individual’s cognition and interest in nature and is characterized by positive feelings of affinity, freedom, safety, and love of nature in the natural environment. They found that this emotional bonding predicted an individual’s environmental protection behavior.

Nisbet et al. explained nature relatedness from multiple perspectives, including an individual’s understanding and identification with relationships that exist between humans and everything else in nature, the love and desire to participate in nature, and the awareness of nature’s importance in all aspects [31]. Noting that nature relatedness is relatively stable over time and context but not set, they developed the Nature Relatedness Scale (NR) and its shorter version (NR-6) to assess differences in individuals’ level of nature connectedness [31,32].

Although there are different terms and definitions of nature connectedness, it is generally considered to be an emotional bond that explains an individual’s understanding of and identification with nature and the quality of the relationship between humans and nature. In this study, we use the terms nature connectedness, the cognitive connection, and emotional bond between humans and nature interchangeably. Research has shown that the more people directly and positively commune with nature, the stronger and more intimate their connection with nature. This phenomenon may be related to the richness of sensory experiences and the positivity of these experiences [27].

With modern technology, it is easy to simulate a natural environment to provide similar sensory experiences, which is socially beneficial. However, Kjellgren and Buhrkall found that a simulated environment did not enhance an individuals’ senses or create the feeling of a harmonious experience [33]. In other words, people do not consider a simulation to be an authentic experience and do not feel like they are part of nature. This suggests that to enhance nature connectedness, nature-based activities must take place in a natural environment. Thus, we propose the following hypothesis:

**Hypothesis** **1 (H1).**
*Nature-based activities significantly increase the level of nature connectedness of stressed individuals.*


### 2.3. State Anxiety

Anxiety has been discussed broadly in the field of psychology. Despite numerous definitions of anxiety, there is a general consensus among researchers that anxiety is an unpleasant emotion. According to the American Psychological Association, anxiety is characterized by feelings of tension, worried thoughts, and physical changes [34]. Inspired by personality psychology, Spielberger suggested that anxiety consists of a state of anxiety and/or a trait (tendency) toward anxiety [35]. Furthermore, state anxiety is a temporary and changeable state triggered by a specific scenario, usually characterized by “I’m feeling nervous now”. The state sub-scale of the State–Trait Anxiety Inventory assesses immediate or recent experiences of dread, fear, and nervousness at a specific time or in a specific situation [36]. Some scholars have argued that anxiety manifests in different dimensions. Ree et al. developed the State–Trait Inventory for Cognitive and Somatic Anxiety (STICSA) to distinguish between cognitive anxiety and somatic anxiety [37]. In this study, we examined both cognitive and somatic anxiety.

Early in the COVID-19 pandemic, studies found that frontline workers experienced high levels of anxiety [2,6,38]. Studies have shown that physical activity in a natural environment reduces anxiety [15,19,39] and nature-based therapies have been broadly used to reduce anxiety; hence, we propose the following hypotheses:

**Hypothesis** **2 (H2).**
*Nature-based activities significantly lower the state anxiety levels of stressed individuals.*


**Hypothesis** **2a (H2a).**
*Nature-based activities significantly lower the cognitive anxiety levels of stressed individuals.*


**Hypothesis** **2b (H2b).**
*Nature-based activities significantly lower the somatic anxiety levels of stressed individuals.*


### 2.4. Perceived Restoration

A restorative environment is one that promotes rejuvenation from the negative emotional state that accompanies mental fatigue and stress [40]. According to attention restoration theory, a restorative environment is characterized by the qualities of fascination, being away, extent, and compatibility [20], and natural environments are restorative and help one to recover directed attention. Harting et al. developed the Perceived Restorativeness Scale to assess an individual’s perceived restoration in a restorative environment [41]. Later, Harting et al. [42] developed the scale further, and Nordh et al. [43] developed a shorter version of the scale.

### 2.5. The Five Senses

The five senses of sight, touch, hearing, smell, and taste are important factors in horticultural therapy, but there is little research on nature-based activities and the healing effects of nature on the human senses. Ulrich found that observing natural scenery had a positive effect on patients’ emotions and recovery [44,45]. Furthermore, individuals prefer the sounds of nature than the sounds of an urban environment [46,47], which help to reduce their stress [48] and anxiety levels [49]. Natural scents, such as the smells of flowers, herbs, and earth after rain, are undeniably refreshing. Research has confirmed the effects of aromatherapy on reducing stress and anxiety [50]. However, few studies have focused on the tactile and gustatory senses, although they may have a considerable effect on well-being. Because multiple sensory experiences are a part of the experiences of nature [51]. Studies have shown that a positive sensory experience may be an important antecedent of the healing effects of nature on humans [52] because multiple sensory experiences are part of the experience of nature [51]. Therefore, we propose the following hypothesis to determine how the stimulation of the five senses affects the levels of perceived restoration and nature connectedness in stressed individuals.

**Hypothesis** **3 (H3).**
*A high level of sensory stimulation (vs. low level) significantly increases (vs. decreases) the perception of restoration.*


**Hypothesis** **4 (H4).**
*A high level of sensory stimulation (vs. low level) significantly increases (vs. decreases) the level of nature connectedness.*


Tang et al. found that individuals with a higher level of nature connectedness experienced greater perceived restoration after experiencing the natural environment and indicated that they were more likely to recover from fatigue and have a greater ability to maintain their own health [52]. Therefore, we propose the following hypothesis to further examine how nature connectedness affects perceived restoration.

**Hypothesis** **5 (H5).**
*A high level of nature connectedness (vs. low level) significantly increases (vs. decreases) the perception of restoration.*


Research has shown that individuals’ level of nature connectedness can be improved by engaging with nature more frequently [53,54], which enhances the emotional bond between humans and nature [55]. Furthermore, sensory experiences reduce individuals’ anxiety [56,57]. Therefore, we propose the following hypothesis.

**Hypothesis** **6 (H6).**
*Nature connectedness mediates the relationship between sensory stimulation and perceived restoration.*


## 3. Materials and Methods

### 3.1. Participants and Procedure

During the COVID-19 pandemic, frontline workers and their families experienced tremendous stress, state anxiety, and sensory attenuation. China’s Nature Education Network, a non-profit alliance of eco-groups and environmentalists with a focus on promoting nature education and nature therapy, conducted a series of nature-based activities called Make a Date with Nature for these workers and their families with the aim of easing their stress levels.

The main purpose of Make a Date with Nature was to help participants relax, recover from stress, and cultivate their intention to continuously connect with nature through guided activities to experience nature through their five senses. The program included sensory activities, such as looking at the night sky; social activities, such as cooperative games; and meditation. In the first part of Make a Date with Nature, the organizers invited participants and explained the purpose of the nature-based activities. After obtaining the participants’ consent, the organizers informed them of the agenda and stayed in touch with them. Small games were played to establish trust 20 min before the start of the nature-based activities. The second part was the nature-based activities, wherein the participants were encouraged to *XunBao*, or explore nature using all of their senses. In the third part, the participants were gathered to meditate and share their experiences of the activities.

We invited the participants of Make a Date with Nature, who were invited by the China Nature Education Network, to voluntarily fill out electronic questionnaires. A pre-activity questionnaire was sent to a group in the messaging app WeChat 10 min before the activities commenced, and a post-activity questionnaire was sent 10 min after the activities ended. At the beginning of each questionnaire, the participants were asked if they were willing to fill out the questionnaire. If they selected “yes”, the page displayed the questionnaire, and if “no” was selected, the page displayed, “Thank you for your participation!” We also required the survey participants to be over 18 years of age. The Make a Date with Nature activities were held at different places in Kunming, Huaibei, and Wuhan between June and December 2020. We collected 283 pairs of pre- and post-activity questionnaires, of which 251 were found to be valid.

### 3.2. Measures

Nature connectedness was measured using the NR-6 scale [32], which contains six items. State anxiety was measured using the STICSA [37], which contains 11 items for cognitive anxiety and 10 items for somatic anxiety. Perceived restoration was measured using a short version of the Perceived Restorativeness Scale [43], which contains five items. Following Franco et al. [50], we developed a 10-item five senses scale to assess sensory stimulation. First, the scales were translated into Chinese by a professional in the field. Next, experts and workers from the China Nature Education Network were invited to revise the wording of the items to ensure that there was no ambiguity. For example, the item “I feel connected to nature when I touch the wind or the water” was changed to “I feel connected to nature when I feel the wind or play with water”. The pre-activity questionnaire assessed the participants’ experience of nature connectedness and anxiety before they took part in the nature-based activities. The post-activity questionnaire assessed their levels of sensory stimulation, nature connectedness, perceived restoration, and anxiety after taking part in the nature-based activities.

All of the items were measured on a 5-point Likert-type scale (1 = *strongly disagree* to 5 = *strongly agree*) except the STICSA. The STICSA model does not contain a neutral option [37]. Hence, our survey used a scale from 1 = *not at all* to 4 = *very much so*. All data were processed using IBM SPSS 22.0 and SmartPLS 3.0.

### 3.3. Measurement Model

We assessed the measurement model before conducting the data analysis. Composite reliability (CR) and Cronbach’s α were used to assess the reliability of the scales. Table 1 shows that the CR values of all of the constructs ranged from 0.947 to 0.988, exceeding the recommended value of 0.70 [58], and Cronbach’s α ranged from 0.929 to 0.986, exceeding the recommended value of 0.70 [59], which indicated that the scales were reliable. Average variance extracted (AVE) values, and factor loadings were used to assess scale validity. Table 1 shows that the AVE values for all of the constructs ranged from 0.755 to 0.890, exceeding the recommended value of 0.50 [58], and the factor loadings ranged from 0.761 to 0.972, exceeding the recommended value of 0.70, showing that the scales had convergent validity. The coefficients of the constructs in Table 2 were smaller than the arithmetic square root of the AVE, confirming the discriminant validity of the scales.

In subsequent analysis, the item scores for sensory stimulation, nature connectedness, and perceived restoration were averaged to calculate the construct scores. Following Ree et al. [37], the item scores for cognitive anxiety and somatic anxiety were added to obtain a total value. The total value was processed using zero-mean normalization to calculate the construct score.

### 3.4. Common Method Bias

Harman’s single-factor test was used to assess common method variance. All of the variables were loaded into an exploratory factor analysis. If multiple factors are obtained and the first factor does not account for more than 40% of the variance, then common method variance is acceptable [60]. The results showed that multiple factors were obtained, and the mutation rate interpretation of the first factor was under 40% (34.89% for the pre-activity questionnaire and 34.43% for the post-activity questionnaire).

## 4. Results

### 4.1. Descriptive Analysis

Two hundred and fifty-one valid questionnaires were used for the descriptive analysis. Most of the participants were women (78.1%) (which is unsurprising because most of the Make a Date with Nature participants were women); 27.9% of the participants were aged 18–30 years, 52.6% were 31–40 years old, and 19.5% were 41–50 years old; 83.7% of the participants were volunteers, who were entrusted with taking the temperature and entering the data in the computer before the COVID test and provided consulting service for residents; 6.8% were medical workers, 1.6% were police officers, and 8.0% were government employees.

### 4.2. Difference after Nature-Based Activities

We conjectured that there would be a difference in the participants’ levels of nature connectedness, cognitive anxiety, and somatic anxiety before and after the nature-based activities. Table 3 shows the means, standard deviations (SD), and standard errors (SEs) of the three variables.

We used a paired sample *t*-test (N > 30) to examine the differences. The results in Table 4 show that the number of participants who showed a decrease in nature connectedness was close to the number of participants who showed an increase in nature connectedness; there was no significant difference in the level of nature connectedness before and after the activities (t = −0.234, df = 250, *p* = 0.816). Thus, H1 was not supported. The number of participants who showed a decrease in cognitive anxiety was greater than the number of participants who showed an increase in cognitive anxiety; there was a significant difference in the level of cognitive anxiety before and after the activities (t = 9.540, df = 250, *p* = 0.000) Thus, H2a was supported. The number of participants who showed a decrease in somatic anxiety was greater than the number of participants who showed an increase in somatic anxiety; there was a significant difference in somatic anxiety before and after the activities (t = 6.109, df = 250, *p* = 0.000). Thus, H2b was supported.

### 4.3. Mediation Analysis

We conjectured that nature connectedness would mediate the effect of sensory stimulation on perceived restoration. We used Hayes’ [61] PROCESS Model 4 with 5000 bootstrap samples to analyze the data from the post-activity questionnaire to examine the mediating effect (Figure 1).

The results revealed that sensory stimulation had a positive and significant effect on perceived restoration (B = 0.247, β = 0.298, SE = 0.037, *p* = 0.000). Thus, H3 was supported. Sensory stimulation also had a positive and significant effect on nature connectedness (B = 0.556, β = 0.632, SE = 0.043, *p* = 0.000), which supported H4. Nature connectedness was found to be positively associated with perceived restoration (B = 0.586, β = 0.622, SE = 0.042, *p* = 0.000), which supported H5. We also found that sensory stimulation had a positive and significant indirect effect on perceived restoration (effect = 0.569). The 95% confidence interval (0.172, 0.485) did not include 0, which indicated that the mediating effect of nature connectedness was significant. Thus, H6 was supported.

## 5. Discussion

Alternative therapies use natural elements to promote physical and psychological health. Mental health practitioners believe that nature and nature-based activities have a healing effect. This study explored the effect of nature-based activities on essential workers during the COVID-19 pandemic and investigated whether nature-based activities affected their perceived restoration through sensory stimulation and nature connectedness.

We found that participating in nature-based activities significantly decreased stress, especially cognitive anxiety and somatic anxiety, which indicates that such activities are psychologically beneficial for people experiencing high levels of stress. Nature-based activities are held in the natural environment. According to the concept of stress recovery, the natural environment is less threatening and makes people feel safe. Therefore, nature-based activities help people recover from stress. However, our results did not support the hypothesis that participating in nature-based activities increases the level of nature connectedness. Nisbet et al. suggested that although attraction to nature and a preference for contact with nature are innate, individual differences remain [31]. Nature connectedness can be regarded as a trait-like emotional bond affected by inherent factors, such as personality, and acquired factors, such as continuous training. Changing the level of nature connectedness may require systematic training over time. Therefore, short-term and discontinuous nature-based activities may not have an effect on individuals’ nature connectedness. There are no specific requirements for nature-based activities; people should engage all of their senses in nature and explore nature freely. In this process, people are in an undirected attention mode, thereby obtaining perceived restoration. We found that the effect of nature-based activities on stressed individuals’ perceived restoration through its direct and indirect effects on sensory stimulation played a mediating role in cultivating nature connectedness. The direct effect of sensory stimulation supports the argument of Oh et al. that the five senses are the antecedents of the positive healing effect of nature-based activities [51]. The mediating effect also supports the positive effect of nature connectedness on improved cognition.

This study examined the positive effect of nature-based activities on reducing anxiety and improving health restoration among workers exposed to COVID-19-related stress. Frontline workers in epidemic prevention and control activities should be given more attention, as these activities are ongoing and becoming part of normal life. Our findings suggest that social welfare programs, such as the China Nature Education Network’s Make a Date with Nature, provide real benefits to people managing the COVID-19 pandemic. Tourist destinations with natural attractions and providers of nature-based activities could collaborate on a non-profit project for frontline workers similar to the one provided by the China Nature Education Network. Our findings suggest that sensory stimulation plays an important role in nature-based activities; hence, we recommend that government departments involved in epidemic prevention and control publish a handbook for frontline workers, teaching them how to manage their stress levels by engaging their five senses while interacting with nature.

It is worthwhile to consider the limitations of this study. We used a within-group test to examine changes in the levels of nature connectedness and state anxiety of individuals experiencing stress due to their pandemic-related work. However, to meet the requirements of the organizers of Make a Date with Nature, we had to use convenience sampling and collect data in a short time, so we were unable to rigorously control for participant selection. Future studies could consider stricter participant requirements. Additionally, a mixed design with within- and between-group tests could be considered to examine the positive effects of nature-based activities more deeply. A within-group design could be used to reveal changes in individuals, and a between-group design could reveal differences between people who participate in nature-based activities and those who do not. Wang et al. indicated that anxiety, depression, and stress are widespread [62]. Further research is needed to investigate whether our conclusions can be generalized to other groups of stressed people. Future studies should carefully consider their sampling methods to further explore the effect of nature-based activities on nature connectedness. Moreover, it should be noted that Make a Date with Nature is held in many cities and at different times of the year, and while the form and content of the events are similar, there may be differences due to differences of time and environment.

Howell et al. suggested that mental health is connected to the feeling of nature connectedness [63]. We recommend that scholars and medical professionals pay more attention to the psychological factors of well-being, self-identification, and self-achievement and harness the recuperative health benefits of the “close-to-nature activities” highlighted in this study to design future recuperation programs for other vulnerable groups, such as stressed workers, students, and senior citizens.

## 6. Conclusions

The results of this study showed that nature-based activities have a healing effect on psychological health, alleviate state anxiety, and improve the feeling of nature connectedness and perceived restoration. Future research could further explore the relationship between nature-based activities and other psychological factors.

## Figures and Tables

**Figure 1 ijerph-19-16501-f001:**
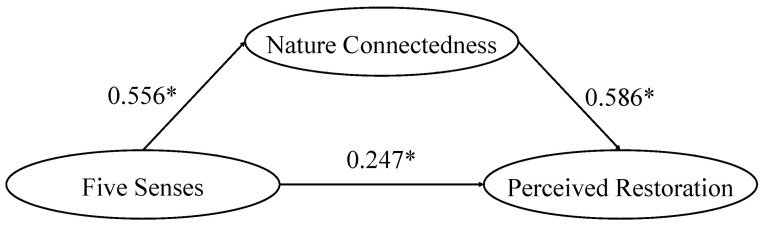
Mediating Effect of Nature Connectedness. * *p* < 0.05.

**Table 1 ijerph-19-16501-t001:** Measurement statistics of the construct scales.

Construct	Item	Factor Loading	Cronbach’s Alpha	Composite Reliability	Average Variance Extracted
Five Senses	FS 1	0.961	0.986	0.988	0.890
	FS 2	0.957			
	FS 3	0.933			
	FS 4	0.932			
	FS 5	0.923			
	FS 6	0.971			
	FS 7	0.962			
	FS 8	0.972			
	FS 9	0.912			
	FS 10	0.920			
Nature Connectedness	NC 1	0.761	0.934	0.947	0.780
	NC 2	0.837			
	NC 3	0.930			
	NC 4	0.874			
	NC 5	0.905			
	NC 6	0.895			
Perceived Restoration	PR 1	0.900	0.929	0.948	0.755
	PR 2	0.816			
	PR 3	0.941			
	PR 4	0.923			
	PR 5	0.830			

**Table 2 ijerph-19-16501-t002:** Discriminant validity of the constructs.

	Five Senses	Nature Connectedness	Perceived Restoration
**Five Senses**	**0.943**		
**Nature Connectedness**	0.635	**0.869**	
**Perceived Restoration**	0.697	0.817	**0.883**

**Table 3 ijerph-19-16501-t003:** Descriptive analysis.

	Nature Connectedness (pre)	Nature Connectedness (post)	Cognitive Anxiety (pre)	Cognitive Anxiety (post)	Somatic Anxiety (pre)	Somatic Anxiety (post)
Mean	4.24	4.25	19.84	16.31	13.75	12.03
SE	0.04	0.05	0.39	0.36	0.28	0.28
SD	0.70	0.76	6.16	5.76	4.48	4.49

**Table 4 ijerph-19-16501-t004:** Paired sample *t*-test results.

	Paired Differences					t	df	Sig. (2-Tailed)
	Mean	SD	SE	95% Confidence Interval of the Difference				
				Lower	Upper			
Nature Connectedness (pre) - Nature Connectedness (post)	−0.012	0.811	0.052	−0.112	0.089	−0.234	250	0.816
Cognitive Anxiety (pre) - Cognitive Anxiety (post)	3.530	5.862	0.370	2.801	4.259	9.540	250	0.000
Somatic Anxiety (pre) - Somatic Anxiety (post)	1.721	4.463	0.282	1.166	2.276	6.109	250	0.000

## Data Availability

The data are not publicly available due to privacy protection for essential workers.

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
