# Peer review of "How Do Nature-Based Activities Benefit Essential Workers during the COVID-19 Pandemic? The Mediating Effect of Nature Connectedness"

_ijerph, 2022, doi:10.3390/ijerph192416501_

Round 1

Reviewer 1 Report

Explain in detail in Method section, especially regarding how you determined the sample size, how about the validity of instrument? How you minimized the bias of the data? 

Author Response

Dear Professor, thank you very much for giving us the opportunity to revise and resubmit our manuscript ("How Do Nature-based Activities Benefit Essential Workers During the COVID-19 Pandemic? The Mediating Effect of Nature Connectedness ").

We greatly appreciate the constructive and insightful comments from you. These comments are especially helpful for use to revise the manuscript. We have seriously considered these comments and have made every effort to address the comments in the revision. We hope that you find the revised manuscript much improved. We will be glad to make further improvements if you have any additional suggestions.

Reviewer 2 Report

The study does not as proposed explain the underlying mechanism of the nature-based activities' healing effect. The study does not demonstrate how Attention restoration theory [20, 21], and the concept of stress recovery [22], manifested in the attained results. The explanation of how the term "connection with nature" was used in this study as well as the description of the procedure of a series of nature-based activities called Make a Date with Nature is not specified and is unclear. Should be a mistake in the report of results (rows 306, 306): "The number of participants whose cognitive anxiety  decreased was greater than the number whose cognitive anxiety decreased.." Overall the study is very interesting and research in this direction is needed. I recommend making clarifications in the text of the article.

Author Response

(The authors gave the same response as above.)

Reviewer 3 Report

Authors have presented a theoretical framework entitled “How Have Nature-based Activities Benefited Essential Workers During the COVID-19 Pandemic? The Mediating Effect of Nature Connectedness” to understand the effect of nature-based activities in healing of psychological problems like stress which is experienced by the essential COVID-19 workers who participated in epidemic prevention and control and their family members were recruited as subjects for this study. Though this type of questionnaire-based study to analyse nature-connectedness is not novel, the authors’ vivid explanation regarding the psychological effects of nature connectedness on the frontline workers of COVID-19 pandemic is quite satisfactory.

The authors should clearly declare the limitations of their study before provide generalized conclusions.

Also, it is needed to conclude more reliable conclusions about their study and possible steps to be avoided in the critical situation of COVID-19 pandemic.

The authors should gather more participants from different socio-economic levels since study based on only 251 valid questionnaires can’t convey generalized results especially in terrible situation of COVID-19 pandemic.

The quality of the manuscript is overall good, although it is needed to moderately improve the English language and style of the manuscript. Also there’s some typos and grammatical errors have to be extracted.

I am strongly recommending that the manuscript may be accepted for publication after major revision.

Author Response

(The authors gave the same response as above.)

Round 2

Reviewer 3 Report

Read the manuscript carefully and correct some spelling and grammar error.

I am Recommending that manuscript may be accept.